Neck circumference is a highly reliable anthropometric measure in older adults requiring long-term care

Sato Ryo 1
http://orcid.org/0000-0001-5679-2391 Sawaya Yohei 1 2
http://orcid.org/0000-0001-6115-1900 Ishizaka Masahiro 2 ishizaka@iuhw.ac.jp
Yin Lu 1
Shiba Takahiro 1
Hirose Tamaki 1 2
http://orcid.org/0000-0002-1771-024X Urano Tomohiko 1 3 turanotky@gmail.com
1 Department of Day Rehabilitation, Care Facility for the Elderly “Maronie-en”, Nishinasuno General Home Care Center , Nasushiobara, Tochigi , Japan
2 Department of Physical Therapy, School of Health Sciences, International University of Health and Welfare , Otawara, Tochigi , Japan
3 Department of Geriatric Medicine, School of Medicine, International University of Health and Welfare , Narita, Chiba , Japan
Gray Andrew
Electronic publication date: 2024 Jan 31
Publication date: 2024
Volume: 12
Electronic Location ID: e16816
Received 2023 Jun 21; Accepted 2023 Dec 29
Copyright: © 2024 Sato et al.
Copyright year: 2024
Copyright holder: Sato et al.
License: This is an open access article distributed under the terms of the Creative Commons Attribution License, which permits unrestricted use, distribution, reproduction and adaptation in any medium and for any purpose provided that it is properly attributed. For attribution, the original author(s), title, publication source (PeerJ) and either DOI or URL of the article must be cited.
License URL: https://creativecommons.org/licenses/by/4.0/

Keywords: Bioelectric impedance analysis, Reliability, Neck circumference, Long-term care, Edema

Funding: Japan Society for the Promotion of Science Grants-in-Aid for Scientific Research 23K06873, 21K10581, and 22K17539 This work was supported by the Japan Society for the Promotion of Science Grants-in-Aid for Scientific Research (grant numbers 23K06873, 21K10581, and 22K17539). The funders had no role in study design, data collection and analysis, decision to publish, or preparation of the manuscript.

==============================
The reliability of neck circumference measurement as an assessment tool for older adults requiring long-term care remains unknown. This study aimed to evaluate the reliability of neck circumference measurement in older adults requiring long-term care, and the effect of edema on measurement error. Two physical therapists measured the neck circumference. Intraclass correlation coefficient (ICC) and Bland–Altman analyses were performed to examine the reliability of neck circumference measurement. Correlation analysis was used to evaluate the relationship between edema values (extracellular water/total body water) and neck circumference measurement difference. For inter-rater reliability of neck circumference measurement, the overall ICC (2,1) was 0.98. The upper and lower limits of the difference between examiners ranged from −0.9 to 1.2 cm. There was no association between edema values and neck circumference measurement error. Thus, measurement of the neck circumference in older adults requiring long-term care is a reliable assessment tool, with a low error rate, even in older adults with edema.

Introduction

The worldwide population of people aged 60 years or older is expected to double by 2050, with significant health and economic consequences (Shlisky et al., 2017). With the aging of the population, increasing numbers of older adults will require long-term care; accordingly, care and support for older adults are becoming important issues (Abdi et al., 2019). Older adults requiring long-term care have higher rates of malnutrition (Sato et al., 2021). Malnutrition is associated with further decline in physical function (Reber et al., 2019), reduced activities of daily living (ADLs) (Mukundan et al., 2022), and increased mortality (Norman, Haß & Pirlich, 2021). This suggests the importance of early detection of poor nutritional status in older adults through screening, allowing for early intervention (Serón-Arbeloa et al., 2022).

Limb circumference measurement is commonly used as a simple nutritional screening assessment in facilities without specialized equipment (Weng et al., 2018). However, circumferential measurement of the extremities in older adults who require long-term care presents many challenges. For instance, proper evaluation of the circumference of the extremities is difficult in individuals with edema, resulting in overestimation of the circumference of the lower extremities (Ishida et al., 2019), and in those with limb contractures. Among anthropometric indices, measurement of the neck circumference is less time-consuming, and is less susceptible to limb impairment. Several studies have reported that neck circumference is associated with nutritional status (Wakabayashi & Matsushima, 2016) and swallowing-related muscle strength (Yoshida et al., 2019), and is also a predictive marker for sarcopenia (Sato et al., 2022) and frailty (Tanaka et al., 2020). Further, Lardiés-Sánchez et al. (2019) reported that the neck circumference is more accurate for nutritional screening than lower leg and upper arm circumferences in older adults requiring long-term care. This suggests that the neck circumference is a useful nutritional screening tool for older adults requiring long-term care.

Anthropometric methods have inherent limitations due to measurement error. Inter-rater reliability must be verified to clarify the clinical utility of anthropometry. However, the reliability of neck circumference measurement in older adults has not been clarified (LaBerge et al., 2009; Stomfai et al., 2011). We hypothesized that the neck circumference measurement as an assessment tool for older adults requiring long-term care is highly reliable. Therefore, this study aimed to clarify (1) the examiner reliability of neck circumference measurement in older adults requiring long-term care, and (2) the effect of edema on the measurement error in neck circumference.

Materials and Methods

Study design

We conducted a single-site cross-sectional study, as an all-participant survey, in March 2023. The Ethical Review Committee of the International University of Health and Welfare approved this study (approval numbers: 21-Io-22-2, 17-Io-189-7), and all participants (or their family members) provided signed informed consent. The study was conducted in accordance with the principles of the Declaration of Helsinki.

Study setting and participants

Participants comprised older adults who were determined to be in need of support or care under the Japanese long-term care insurance system and used a day care rehabilitation facility for older adults (Yamada & Arai, 2020). All 146 participants using day care facility were approached in March 2023, and all users were able to give consent. Only 108 older adults were included in the study analysis. Exclusion criteria were as follows: (1) age under 65 years, (2) age over 100 years, (3) inability to measure extracellular water to total body water (ECW/TBW), and (4) missing data (Fig. 1).

Figure 1 Flowchart of study participant selection.

ECW/TBW: extracellular water/total body water.

Data collection and measures

Neck circumference

Two physiotherapists working at the facility (Examiner A and Examiner B) each measured the neck circumference of the participants. Neck circumference was measured in millimeters using a measuring tape, in accordance with a previous study of older adults requiring long-term care in Japan. Specifically, the neck circumference was measured perpendicular to the longitudinal axis of the neck, just below the larynx (thyroid cartilage), with the participant in the sitting position (Wakabayashi & Matsushima, 2016). Prior to the physical measurement evaluation, we ensured that there were no differences in measurement methods between the examiners. The two examiners measured the same participant twice, in a blinded fashion so that the other examiner would not know the results of the measurement.

Covariates

Edema was assessed as the ECW/TBW using a body composition analyzer (InBodyS10; InBody, Tokyo, Japan). ECW/TBW is higher in cases with increased fluid, and ECW/TBW, as measured by bioelectrical impedance analysis, is associated with the prognosis in older adults (Inal et al., 2014; Zheng et al., 2022). ECW/TBW ≥ 0.400 is considered to reflect overhydration; thus, in accordance with previous studies, participants with ECW/TBW ≥ 0.400 were considered as overhydrated (Zheng et al., 2022). Age, sex, height, pre-existing medical conditions, and ADL as assessed by the Barthel Index, were obtained from the facility’s medical records (Bouwstra et al., 2019). Data on cerebrovascular diseases, orthopedic conditions, intractable neurological diseases, cancer, cardiovascular diseases, respiratory diseases, diabetes, and hypertension were obtained from institutional medical records. Pre-existing medical conditions were defined as physician diagnoses.

Statistical analysis

Intraclass correlation coefficient (ICC), and Bland–Altman analyses were performed to evaluate the reliability of neck circumference measurement. To test the intra-examiner reliability of the two examiners, ICC (1,1) was performed for Examiner A and Examiner B separately. For inter-rater reliability, ICC (2,1) was calculated using the first value of the two neck circumference measurements for each examinee. ICC values below 0.50 indicate low reliability, values between 0.50 and 0.75 indicate moderate reliability, values between 0.75 and 0.90 indicate good reliability, and values greater than 0.90 indicate high reliability (Koo & Li, 2016).

Bland–Altman analysis was used to examine the types of bias among examiners. A scatterplot (Bland–Altman plot) was created with the difference between the two measurements (value of Examiner A—value of Examiner B) (d) on the y-axis and the mean of the two measurements on the x-axis, and the limit of agreement (LOA) was calculated (Bland & Altman, 1986).

In addition, Pearson’s correlation analysis was used to evaluate the relationship between the ECW/TBW and inter-rater measurement absolute value of the difference.

Results

A total of 108 older adults (66 men and 42 women) participated in this study (Fig. 1). Table 1 summarizes the basic attributes and pre-existing medical conditions of the study participants.

Table 1 Participant characteristics.

	Total (n = 108)	
Sex, female (%)	38.9	
Age (years)	79.9 ± 6.5	
Height (cm)	159.9 ± 9.0	
Weight (kg)	59.3 ± 10.7	
BMI (kg/m2)	23.2 ± 3.5	
Barthel index (score)	84.3 ± 18.4	
Neck circumference (cm) Examiner A first	36.5 ± 2.9	
Examiner A second	36.5 ± 2.9	
Examiner B first	36.4 ± 3.0	
Examiner B second	36.4 ± 3.0	
ECW/TBW	0.4 ± 0.0	
SMI (kg/m2)	6.4 ± 1.1	
FAT (%)	28.7 ± 9.9	
Cerebrovascular disease (%)	60.2	
Orthopedic disease (%)	54.6	
Intractable neurological disease (%)	13.0	
Cancer (%)	17.6	
Cardiovascular disease (%)	25.9	
Respiratory disease (%)	14.8	
Diabetes mellitus (%)	23.1	
Hypertension (%)	46.3	
Note:

Data are presented as the mean ± standard deviation, unless otherwise noted. BMI, body mass index; ECW/TBW, extracellular water/total body water; SMI, skeletal muscle mass index; FAT, body fat percentage.

The intra-rater reliability for neck circumference measurement was high, with ICC (1,1) >0.99, for both examiners A and B (Table S1). The inter-rater reliabilities for neck circumference measurement are shown in Table 2. The inter-rater reliability for neck circumference measurement in all participants was high, with ICC (2,1) = 0.98. The inter-measurement bias was 0.1 cm (95% LOA: −0.9 to 1.2 cm) (Fig. 2). Among participants with ECW/TBW ≥ 0.400 (edematous group), the inter-rater reliability for neck circumference was also high, with ICC (2,1) = 0.98. In addition, the inter-measurement bias was 0.2 cm (95% LOA: −0.8 to 1.1 cm) (Fig. 2).

Table 2 Inter-examiner reliability for neck circumference measurement.

	n	ICC (2,1)	95% CI	Bias	95% LOA	
ALL	108	0.98	[0.97–0.99]	0.1	−0.9 to 1.2	
Male	66	0.98	[0.97–0.99]	0.0	−0.9 to 1.0	
Female	42	0.95	[0.88–0.98]	0.3	−0.8 to 1.4	
Age (years) 65–74	19	0.98	[0.94–0.99]	0.1	−1.2 to 1.3	
Age (years) ≥ 75	89	0.98	[0.98–0.99]	0.1	−0.8 to 1.1	
ECW/TBW < 0.40	68	0.98	[0.97–0.99]	0.1	−1.0 to 1.2	
ECW/TBW ≥ 0.40	40	0.98	[0.96–0.99]	0.2	−0.8 to 1.1	
Note:

ICC, intraclass correlation coefficient; LOA, limit of agreement; MDC, minimal detectable change; CI, confidence interval; ECW/TBW, extracellular water/total body water; Bias, mean value of the difference between methods.

Figure 2 Bland-Altman plots for the inter-rater reliability of neck circumference measurement.

Bland-Altman plots are shown for (A) all participants (bias: 0.1, 95% LOA: −0.9 to 1.2) and (B) the edematous group (bias: 0.2, 95% LOA: −0.8 to 1.1). Bias reflects the mean value of the difference between examiners, and the 95% LOA (limit of agreement) was calculated as follows: bias ± 1.96 × (standard deviation of the difference between examiners). The middle line denotes the bias and dashed lines denote the 95% LOA.

No association was found between the ECW/TBW and inter-rater measurement absolute value of the difference (r = −0.01, p = 0.951) (Fig. 3).

Figure 3 Relationship between ECW/TBW and absolute differences in examiner measurements.

No association was found between the ECW/TBW and inter-rater differences in measurement. ECW/TBW: extracellular water/total body water. Straight lines indicate approximate curves.

Discussion

The present study evaluated the inter-rater reliability of neck circumference measurement in older adults requiring long-term care. We found that the inter-rater reliability for neck circumference measurement was ICC (2,1) = 0.98, indicating a high degree of reliability. Furthermore, differences within −0.9 to 1.2 cm in neck circumference values were shown to be acceptable as inter-examiner error. To our knowledge, this study is the first to validate the reliability of neck circumference measurement in older adults requiring long-term care.

Previous studies have shown that BMI is associated with neck circumference and is a screening tool for metabolic syndrome (Hingorjo, Qureshi & Mehdi, 2012; Kroll et al., 2017; Kim, Moon & Yun, 2021). Previous studies have also shown that a decrease in neck circumference is independently associated with sarcopenia (Sato et al., 2022). Neck circumference is also used as an indicator to assess non-communicable diseases, and it has been reported that increased neck circumference is associated with an increased risk of cardiovascular disease (Asil et al., 2021; Caro et al., 2019). While nutritional indices such as BMI may lead to overestimation of sarcopenia due to potential internal edema, Do, Kim & Kang (2022) reported that neck circumference predicts sarcopenia better than BMI in with predicted edema, such as in the case of patients undergoing peritoneal dialysis. Furthermore, Dogan, Ayhan & de Almeida (2023) found that neck circumference can predict sarcopenia as accurately as calf circumference—the standard screening tool for sarcopenia. This suggests that neck circumference may be a useful tool to assess sarcopenia among older people requiring long-term care.

Upper arm and calf circumferences are commonly used circumference measurements. A study of community-dwelling older adults by Foroughi et al. (2011) reported a high degree of concordance for upper arm circumference measurement, with an inter-examiner correlations (r) ranging 0.84 to 0.94. In addition, Jamaiyah et al. (2008) reported a 95% LOA of −0.9 to 0.4 cm and ICC of 0.998 for lower calf circumference measurement in adults aged 60 years or older, indicating high inter-rater reliability. Similar to these previous studies of circumference measurement, the present study results indicate neck circumference as a physical measurement index with high inter-rater reliability. Therefore, the results suggest that neck circumference can be used as a reliable tool in older adults requiring long-term care, even in those with contractures or trauma to the upper and lower extremities or who have difficulty dressing and undressing. Furthermore, the study confirmed high reliability in overhydrated older adults (ECW/TBW ≥ 0.400) and found no association between ECW/TBW and inter-rater differences. This suggests that the neck circumference measurement method is a reliable assessment tool for older adults suffering from a variety of conditions that cause excess water retention. This study reveals, for the first time, the intra- and inter-rater reliability of neck circumference to assess sarcopenia in older adults.

The present study has some limitations. First, this was a single-center study; further multicenter studies are needed for increased generalizability. Second, the measurements were taken by two experts working at the facility; therefore, the procedures used in this study cannot be used in a design in which three or more examiners measure the same participant. Furthermore, reliability may be reduced with larger numbers of examiners. Third, although the two examiners were blinded about each other’s measurements taken in this study—the order of which was also randomized—no counterbalancing was adopted. In the future, it would be desirable to adopt counterbalancing across the study sample. Finally, this study focused on a statistical tool; the final decision regarding the clinical application of the errors identified in this study is left to the researcher/clinician.

Conclusions

The present study found that neck circumference measurement shows high inter- and intra-examiner reliability in older adults requiring long-term care. It was also found to be reliable in those who presented with edema.

Supplemental Information

Supplemental Information 1 Intra-rater reliability for neck circumference measurement.

ICC, intraclass correlation coefficient; CI, confidence interval; ECW/TBW, extracellular water/total body water.

Click here for additional data file.

Supplemental Information 2 Raw data.

Click here for additional data file.

We are grateful to all participants from the Nishinasuno General Home Care Center.

Additional Information and Declarations

Competing Interests

Author Contributions

Human Ethics

Data Availability

The authors declare that they have no competing interests.

Ryo Sato conceived and designed the experiments, performed the experiments, analyzed the data, prepared figures and/or tables, authored or reviewed drafts of the article, and approved the final draft.

Yohei Sawaya conceived and designed the experiments, performed the experiments, analyzed the data, authored or reviewed drafts of the article, and approved the final draft.

Masahiro Ishizaka conceived and designed the experiments, performed the experiments, analyzed the data, authored or reviewed drafts of the article, and approved the final draft.

Lu Yin conceived and designed the experiments, performed the experiments, authored or reviewed drafts of the article, and approved the final draft.

Takahiro Shiba conceived and designed the experiments, performed the experiments, authored or reviewed drafts of the article, and approved the final draft.

Tamaki Hirose conceived and designed the experiments, performed the experiments, authored or reviewed drafts of the article, and approved the final draft.

Tomohiko Urano conceived and designed the experiments, performed the experiments, analyzed the data, authored or reviewed drafts of the article, and approved the final draft.

The following information was supplied relating to ethical approvals (i.e., approving body and any reference numbers):

The study protocol was approved by International University Health and Welfare Ethics Review Board (approval numbers: 21-Io-22-2, 17-Io-189-7).

The following information was supplied regarding data availability:

The raw measurements are available in the Supplemental File.

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
