# Peer review of "Neck circumference is a highly reliable anthropometric measure in older adults requiring long-term care"

_PeerJ, doi:10.7717/peerj.16816_

## Round 0.1 · original submission · Major Revisions

Thank you for your patience. I’m not the original editor of your manuscript but was invited to take that role recently. I’ve read over our three reviewer reports and as you can see, Reviewer #1 has no changes to suggest. Reviewers #2 and #3 have both provided some helpful comments that I think will assist you in revising your manuscript.

Reviewer #2 makes a good point about the reliability versus the utility of the measure. Your title makes it clear to me that you are interested in the reliability of the measure but I agree that you could more clearly motivate why measurements of neck circumference are important/useful in the first instance (expanding on Lines 57–71). Their suggestion that you could consider the validity of this measure by correlating it to body fat (and muscle mass if available) is something you could consider adding, or you could keep your focus entirely on the reliability aspect.

Reviewer #3 suggests a reference to add. You are welcome to include this if you find it helpful. The topic of how valid measurements are against a gold standard would be interesting, but I suspect that it would fall outside of your intentions in the present manuscript. Their other comments warrant your consideration and response.

I will add some of my own comments. These are often requests for clarification of some details of the analyses (speaking as a biostatistician who often works in anthropometry). I look forward to seeing a revised version of your manuscript with responses to each of the reviewers’ and my points and any changes in your manuscript tracked.

Your use of Bland-Altman for two examiners seems slightly unusual here to me. Bland-Altman is for method comparison rather than reliability analysis per se, and your two examiners are not methods of measurement or, as far as I could tell, of specific interest in themselves. Bias between the two examiners would not seem to be of particular interest to me (unless one was, say, experienced and one a novice, which is not the case from Line 100, although exploring such a difference would require more instances of each type, but this would lead to a situation where Bland-Altman analysis would seem entirely appropriate to me). If there was statistically significant evidence of a bias (e.g., Lines 133–135), what would this mean unless we knew exactly who the two raters were (the sign of the bias cannot be interpreted without this information)? If you mean that these are two randomly selected raters, the bias doesn’t seem as interpretable to me and in that case, it is only the greater (in magnitude) value in the LoA that seems particularly important to me. I wonder what you feel the bias and LoA add on top of the ICC.

I would also have considered using a mixed model for the calculation of the LoA to reflect the randomly selected (representative) raters if this was retained but it was not clear to me that this was done. You may well have another interpretation intended here that I’m not picking up on, so could you clearly explain what you’re aiming to achieve with the Bland-Altman analysis in your manuscript beyond what is already shown through the ICC analysis?

I was initially unclear how the ICC(1,1) calculation (Line 125) was performed as each examiner measured each participant twice, giving four measurements per participant. For this, the underlying mixed model is slightly more complicated than with a single assessor. Could you clearly explain what was done around Line 125? From Lines 147–148, it sounds as if this was done separately for each rater, but again without knowing the raters, I find this difficult to interpret and a single intra-rater ICC could be provided using data from both.

I also wondered if it would be useful to use Spearman-Brown’s prophecy formula to indicate how many measurements would be needed from an examiner to achieve satisfactory reliability (e.g., while a single weight or height measurement ought easily to be sufficient for WHO’s 0.95 reliability criterion, multiple circumference measurements can be needed). Given the results you use, your ICCs are high enough to justify single measurements, but you could do this using the lower 95% CI limit for the ICCs (unless these are also greater than 0.95 or whatever you consider acceptable in this specific context).

Related to the above, 95% CIs are needed for all ICCs.

The presentation of Pearson’s correlation on Line 150 (and again on Line 154) should be removed unless it can be justified. Same for Table 2 and anywhere else it is presented. The use of ICCs (and Bland-Altman’s method) is intended to avoid the problems that arise from using such correlation coefficients.

The MDC on Line 152 needs units (cm). The interpretation on Line 166 isn’t quite correct and should be revised. The focus from the title is on reliability, and while reliability affects MDC, I wonder if the presentation of the MDC could be more clearly motivated for the reader.

Were the two measurements taken in random orders for each participant (i.e., was each examiner first for around 52 participants, or, alternatively, were the measurements counterbalanced)?

I think you can look at ECW/TBW against the difference between examiners, but I wouldn’t call that difference “measurement error” (e.g., Lines 38, 142, and 157) as we have no gold standard. Perhaps just “measurement difference”? You could also argue for using the absolute value of this difference if the question was would edema make measurements harder.

I wonder if reporting ICCs for some subgroups, including perhaps categories defined by ECW/TBW such as Line 115, might be interesting and useful. I appreciate that this becomes challenging with the total sample size, but was (and you could do this for both intra- and inter-rater) reliability higher or lower for men or women, with edema (as you give on Line 153 and Table 2 for inter-rater reliability) or without, younger or older? These groups need to be meaningful to you and I mean these only as examples.

When you report descriptives in Table 1, which rater/measurement occasion are these from for neck circumference?

For Figure 3, I would have thought that the difference in measurements (first measurement or mean of both for each rater) would have been on the y-axis.

·

Basic reporting

Clear and unambiguous, professional English used throughout.
Literature references, sufficient field background/context provided.
Professional article structure, tables.
The results regarding the hypotheses are sufficient.

Experimental design

Original primary research within Aims and Scope of the journal.
Method is sufficient.

Validity of the findings

Conclusions are well stated, linked to the .original research question.

Additional comments

This study is important because it provides preliminary information on taking many precautions, including emergency interventions, in determining the severity of edema formed in long-term hospitalized patients.
On the other hand, it is very important for practical evaluation to determine that the inter-rater reliability of the neck circumference measurement in older adults requiring long-term care is very high.

Reviewer 2 ·

Basic reporting

The English language used is scientific, clear and free from language mistakes. The article is well written and constructed. However the hypothesis is not well constructed. The paper only uses data to assess whether inter examiner reliability when measuring neck circumference but does not in any way report on how they came to a conclusion that neck circumference came to a conclusion that neck circumference can be used an assessment tool for older adults. No cut-offs are suggested (even if only in their discussion.) Furthermore the literature is abundant about the use of neck circumference as a proxy measure for non communicable diseases, and this was not touched upon in this paper. Also since body fat was measured and I would think muscle mass, it would be useful to assess whether neck circumference had predictive power for those values.
Finally you always mention that several studies have found et and you only quote one study.

Experimental design

The experimental design is adequate and well explained.

Validity of the findings

The findings need to modified as per earlier comments.

Additional comments

NA

Reviewer 3 ·

Basic reporting

I do not consider myself in a position to evaluate the quality of English because it is not my native language.
I think the literature is sufficient, but it will be important to report the next article https://www.sciencedirect.com/science/article/pii/S0197457222001689 (from the same group of researchers), that evaluate the sarcopenia in the same group of older adults.
The hypotheses is"the neck circumference measurement as an assessment tool for older adults requiring long-term care is highly reliable", but you are testing the precision between two expert working and not the accuracy of the measurement using a gold standard (percent of muscle by body composition analyzer) to evaluate the relevance of the measurement.
There is another article that shown the accurate of neck circumference that can be review https://pubmed.ncbi.nlm.nih.gov/37115681/

Experimental design

The article shows that the performance and rigor of the research is ethical and well developed. But, I concern with the objectives that do not answer the hypothesis. It requires focusing the article on what was done.
The methods described are ok and it is replicable.
Line 93-94: "A total of 108 older adults who met the exclusion criteria were included in the study analysis" I think must be: "who met the inclusion criteria"
There are other anthropometric measures commonly used in older people to assess the nutritional status (weight and height to determine the BMI), which would be good to mention and why they are often not good measures (not juts limb circumferences): prostrate patients, edema, etc.
I think it can reinforce the idea of edema and precision of the measures
Line 63: You mention: "neck circumference has the advantage of being easy" I am not agree, because is easier than others, but not exactly easy to measure. It used to have more error for example when we compare with weight.

Validity of the findings

I think the article can be focus on precision of the method an error between two expert working and change the hypothesis, because it is not evaluated in this article. If you want to evaluate the hypothesis
stated, you need to include sarcopenia in the analysis.

---

## Round 0.2 · accepted · Accept

Thank you for your revised version and rebuttal. I am pleased to accept your manuscript. There are some small edits that I think are needed (listed below) but I think you can address these in the preparation of the final version of your work. Congratulations!

Lines 62–63: “Among anthropometric indices, measurement of the neck circumference has the less time-consuming” should perhaps be “Among anthropometric indices, measurement of the neck circumference is less time-consuming”.

Table 2: “Age (years) 75≤” should perhaps be “Age (years) ≤75” (compare with the following two rows).

Figure 3: Shouldn’t the y-axis be “Absolute difference in…”? I’d suggest adding “absolute” before “differences” in the caption as well.

Reviewer 3 ·

Basic reporting

I do not consider myself in a position to evaluate the quality of English because it is not my native language.
The Literature, format, and results meet the standard

Experimental design

The article is well design and methods are sufficient

Validity of the findings

It is a good tool that can be useful, specially in older adults